# Receptor-Mediated Redox Imbalance: An Emerging Clinical Avenue against Aggressive Cancers

**DOI:** 10.3390/biom12121880

**Published:** 2022-12-15

**Authors:** Xiaofeng Dai, Erik W. Thompson, Kostya (Ken) Ostrikov

**Affiliations:** 1Wuxi School of Medicine, Jiangnan University, Wuxi 214122, China; 2School of Biomedical Sciences, Queensland University of Technology (QUT), Brisbane, QLD 4059, Australia; 3Translational Research Institute, Brisbane, QLD 4102, Australia; 4School of Chemistry, Physics and Center for Biomedical Technologies, Queensland University of Technology (QUT), Brisbane, QLD 4000, Australia

**Keywords:** cold atmospheric plasma, redox imbalance, receptor, aggressive cancer

## Abstract

Cancer cells are more vulnerable to abnormal redox fluctuations due to their imbalanced antioxidant system, where cell surface receptors sense stress and trigger intracellular signal relay. As canonical targets of many targeted therapies, cell receptors sensitize the cells to specific drugs. On the other hand, cell target mutations are commonly associated with drug resistance. Thus, exploring effective therapeutics targeting diverse cell receptors may open new clinical avenues against aggressive cancers. This paper uses focused case studies to reveal the intrinsic relationship between the cell receptors of different categories and the primary cancer hallmarks that are associated with the responses to external or internal redox perturbations. Cold atmospheric plasma (CAP) is examined as a promising redox modulation medium and highly selective anti-cancer therapeutic modality featuring dynamically varying receptor targets and minimized drug resistance against aggressive cancers.

## 1. Introduction

Redox imbalance is defined as a disordered balance between the effects of oxidants and antioxidants [1]. Cancer-related malignant transformations disrupt normal cell functioning by intensifying cellular redox processes. Moreover, the functionality of the antioxidant system is often disrupted in malignant cells, rendering cancer cells more vulnerable to redox stress compared to normal cells. These phenomena collectively associate redox imbalance in cancer cells with the key types of programmed cell death, such as apoptosis, paraptosis, autophagic cell death, ferroptosis, necroptosis, and pyroptosis [2]. Importantly, the fate of redox imbalance-stressed cells is determined by the concentration of reactive oxygen species (ROS), and by the dominant pathways activated in response to the external stress-related stimuli. Indeed, while low ROS levels stimulate normal cell functioning, high levels of ROS typically cause acute cytotoxicity [1].

Cell surface receptors make cancer cells sensitive to various external stimuli. Many targeted therapies and immunotherapies fail due to the evolved mutations in the targeted receptors, triggering the development of dual-targeted strategies such as the combination of venetoclax and rituximab in the treatment of refractory chronic lymphocytic leukemia [3,4]. Despite their great promise, dual-targeting approaches still rely on a limited number of proven receptor-mediated signaling mechanisms and do not represent the ultimate option for cancer cure. The search for new onco-therapeutics based on controlling receptor-mediated redox balance in the cells is thus imperative and timely.

Among the various tools capable of manipulating cellular redox levels, cold atmospheric plasma (CAP), being a fourth state of matter, represents an emerging onco-therapeutic modality and fulfills these requirements, with its selectivity against cancer cells versus normal cells having been demonstrated in multiple types of cancers [5,6,7,8,9]. Specifically, CAP has been shown capable of attenuating the growth of triple negative breast cancer (TNBC) cells [5], halting the migrative ability of TNBC and bladder cancer cells [6], boosting cellular immune response and enabling checkpoint blockade therapy against melanoma [7], enhancing the uptake by and maturation of peripheral blood monocyte-derived dendritic cells together with an inflammatory secretion profile in peritoneal carcinomatosis [9], and reducing the metabolism of chondrosoma cells [8]. This selectivity owes to the unique mechanisms of CAP in disrupting the antioxidant system of malignant cells that cannot be achieved by normal chemical mixtures of the reactive species present in cold plasmas [5] and may be linked to the differential contributions of its cocktail content. 

Through classifying cell receptors by the cancer hallmarks they are primarily associated with and highlighting the main interactions between the ROS and these receptors, we use a focused example of CAP as a highly selective anti-cancer modality. This approach allows us to specify how the plasma-related receptor-mediated redox imbalance effects can contribute to the development of targeted clinical procedures against aggressive cancers (Figure 1). We argue that CAP may cause good selectivity against cancer cells by targeting multiple and dynamically varying targets depending on the plasma dose (among other factors), thereby potentially avoiding the common drug resistance issue. This paper concludes with an analysis of the issues that need to be resolved along the way towards the clinical translation of receptor-mediated, exposure dose-dependent plasma-based therapies against aggressive cancers. 

## 2. Receptors Associated with Redox Imbalance

### 2.1. Growth Factor Receptors: Reprogrammed Cell Proliferation and Migration 

Growth factor receptors are transmembrane proteins that transduce signals to the intracellular space by binding to specific growth factors. There exist diversified types of growth factor receptors, such as epidermal growth factor receptor (EGFR), transforming growth factor β receptor (TGFβR), vascular endothelial growth factor receptor (VEGFR), and platelet-derived growth factor receptor (PDGFR), each of which represents a family of receptors instead of a single molecule. While EGFR and PDGFR family members are mainly responsible for cell proliferation, TGFβR and VEGFR members are primarily involved in cell migration. We select and focus on EGFR and TGFβR here to illustrate how growth factors regulate cancer cell proliferation and migration through redox cycling.

#### 2.1.1. EGFR-Mediated Redox Cycling Plays Dual Roles in Cell Proliferation

The EGFR family contains four members: EGFR (aka human epidermal growth factor receptor (HER)1), HER2, HER3, and HER4 [11]. Each mature EGFR contains an extracellular domain containing the ligand-binding region, a single transmembrane region, and an intracellular tyrosine kinase domain for signal relay. There exist at least 12 different growth factors capable of binding EGFR members, such as EGF, transforming growth factor α (TGFα), amphiregulin, betacellulin, and neuregulins. These growth factors can trigger homo- and/or heterodimerization of EGFR that results in trans-autophosphorylation and subsequent activation of SH2 domain-dependent downstream signaling [12]. Below, we take EGFR as the representative receptor of this family and focus on its roles in redox-mediated cell proliferation.

EGFR is mainly responsible for cell proliferation, with both promotive and suppressive roles being reported. For example, it has been deciphered that benzoapyrene, a known mammary carcinogen in rodents, enhanced breast cancer cell growth by generating hydrogen peroxide (H_2_O_2_) and activating EGFR [13]. Pyrroloquinoline quinone, a redox cofactor for bacterial dehydrogenases, has been implicated as an important nutrient in mammals that can stimulate epithelial cell proliferation by activating EGFR through redox cycling [14]. A copper chelate induced T-cell acute lymphoblastic leukemia cell apoptosis and overcame the multidrug resistance via EGFR/AKT blockage [15]. Oxidative stress has been demonstrated to affect retinal pigment epithelial cell survival via EGFR/AKT signaling [16]. On the other hand, EGFR and Src kinases were documented to promote oxidative stress-dependent apoptosis and loss of cell adhesion in epithelial cells [17]. Besides cell life or death, EGFR is also involved in other processes such as epithelial-to-mesenchymal transition (EMT) and inflammation with, mostly, its tumor-promotive role being reported. For instance, redox regulation of EGFR steered hypoxic mammary cell migration towards oxygen [18], angiotensin II induced EMT in renal epithelial cells via reactive oxygen species (ROS)-mediated EGFR stimulation [19], and PM2.5 triggered pro-inflammatory mediator over-secretion from human bronchial epithelial cells via oxidative stree-relayed EGFR activation [20]. 

In light of the prominent and multi-faceted roles of EGFR, therapeutic strategies taking advantage of EGFR signaling have been proposed for cancer treatment. For instance, in vivo delivery of siRNAs targeting EGFR and BRD4 expression by peptide-modified redox responsive PEG-PEI nanoparticles have been developed for treating triple negative breast cancer cells [21], and redox-sensitive thiolated TPGS (GSH redox-sensitive thiolated vitaminE-PEG1000-succinate)-based nanoparticles have been established to target EGFR as a novel lung cancer therapy [22].

#### 2.1.2. TGFβR-Mediated Redox Cycling Accelerates Cell Migration

TGFβR has three isoforms: TβRI, TβRII, and TβRIII. While TβRI and TβRII form a hetero-tetramer that harbors serine/threonine protein kinases in the cytoplasmic domain that are activated after binding of the ligand TGFβ, TβRIII functions as a co-receptor to increase the ligand-receptor interaction affinity without catalytic activity [23,24]. TGFβ is not a single molecule but represents a family of structurally related proteins comprised of TGFβ, activins/inhibins, and bone morphogenic proteins (BMPs). On ligand binding, TGFβR dimerizes to induce serine/threonine phosphorylation followed by the phosphorylation of its intracellular effectors SMADs (referred to as canonical TGFβ/SMAD signaling) or critical intermediates involved in other pathways such as PI3K/AKT/mTOR, MAPK, and Rho-like GTPase signaling (known as TGFβ/non-SMAD pathways) [25]. 

The pro-invasive and pro-metastatic roles of TGFβR are evident in a broad panel of cancers including colorectal cancer cells [26], triple negative breast cancer cells [27], prostate cancer cells [28], gastric cancer cells [29], pancreatic cancer cells [30], nasopharyngeal cancer cells [31], lung adenocarcinoma cells [32], and osteosarcoma cells [33]. The fundamental roles of TGFβ signaling in cancer metastasis has been attributed to its ability to foster cancer stemness, as exemplified by the elevated SOX4 expression in gastric cancers sustained by TGFβ over-activation [34]. This, at least, has partially explained the enhanced drug resistance accompanying increased TGFβRII [35]. Besides the typical role of TGFβ in cancer metastasis, an over-represented TGFβRII level has been associated with many other tumor-relevant traits such as the promotion of cancer cell growth and survival of cancer-associated fibroblasts (CAFs) [36] and enhanced natural killer (NK) cell activity in mice [37]. Importantly, these aforementioned functionalities of TGFβ have been associated with redox regulation. For instance, pathways such as the ROS-NRF2 [38], ROS-PI3K/AKT/mTOR [39], ROS-MAPK [40], and ROS-NFkB/NOX4 [41] signaling have been identified to mediate TGFβ-triggered EMT. Controversially, TGFβ triggered apoptosis in mouse liver cancer cells, breast cancer cells, pulmonary system adenocarcinoma cells, and lymphoma cells via enhancing cellular ROS level [42], suggestive of the double-edged roles of TGFβ in carcinogenesis. 

Onco-therapeutic strategies through TGFβ signaling blockage have been proposed, which include TGFβR inhibitors such as galunisertib, LY3200882, and vactosertib [43,44,45], TGFβ traps such as AVID200 [46], M7824 [47] and luspatercept [48], antibodies neutralizing TGFβ activity such as fresolimumab [49] and LY3022859 [50], antisense oligonucleotides against TGFβ2 mRNA such as trabedersen [51], and ISTH0047 [52].

### 2.2. Toll-like Receptors: Activated Immune Response

Toll-like receptors (TLRs) are type I transmembrane glycoproteins carrying an ectodomain that contains leucine-rich repeats (LRRs) for ligand recognition, an intracellular Toll-interleukin 1 receptor domain (TIR) for signal transduction induction, and a transmembrane domain; they can recognize damage-associated molecular patterns (DAMPs) or pathogen-associated molecular patterns (PAMPs) and participate in immune responses [53]. On binding of a ligand to a TLR, two receptor chains dimerize and recruit adaptor proteins such as myeloid differentiation primary response gene 88 (MyD88) [54], leading to cytokine production and transcription factor activation for the activated immune response.

There exist at least 10 TLRs in humans and mice: TLR1 to TLR10. In mammalian cells, TLR1, TLR2, TLR5, TLR6, and TLR10 are expressed on the cell surface, TLR3, TLR7, TLR8, and TLR9 localize within endosomal compartments, and TLR4 is expressed in both the membrane and intracellular compartments [55]. TLRs are primarily expressed by innate immune cells and participate in the adaptive immune response. TLRs are also present on many cancer cells and play tumor-promotive or suppressive roles, the outcome of which depends on the type of TLR, tumor cell, and immune cells infiltrating the tumor site. 

While most TLRs, such as TLR2, TLR4, TLR5, TLR7, and TLR9, play dual roles in carcinogenesis, TLR1 and TLR3 are largely tumor-suppressive, as demonstrated by induced apoptosis by TLR3 in human non-small-cell lung cancer cells [56]. TLR1 is expressed on dendritic cells (DCs), NK cells, eosinophils, monocytes, neutrophils, and B cells, and TLR3 is primarily expressed in DCs and NK cells. Accordingly, a TLR1 agonist has been designed [57] and applied for treating cancers such as leukemia [58]; oncolytic retrovirus taking advantage of TLR3-dependent apoptosis has been proposed for ovarian cancer treatment [59]; and a synergistic strategy combining a PLGA-particle vaccine carrying a TLR3/RIG-I ligand riboxxim and an immune checkpoint blockade has been proposed as an effective anti-cancer immunotherapy [60]. Despite the relatively evident tumor-suppressive roles of TLR1 and TLR3, controversies have been reported, such as the promoted tumor growth and cisplatin resistance observed in head and neck cancer cells as a result of activated TLR3 signaling [61]. Agonists of other TLRs have also been proposed to synergize with immunotherapeutic agents for cancer treatment. For instance, a TLR7/8 agonist has been combined with ICD amplifiers to eliminate solid tumors [62]. 

#### 2.2.1. Redox-Dependent TLR Control on Immune Response via Cytokine Production

Various T cell cohorts, including T helper cell type 1 (Th1) and type 2 (Th2), are involved in an adaptive immune response that controls the switch between cellular and humoral immune responses. Specifically, Th1 cells secrete pro-inflammatory cytokines such as interferon gamma (IFNγ) and interleukin 2 (IL2) to stimulate cytotoxic T cells, NK cells, and macrophages and switch on cellular immune responses, and Th2 cells release IL4/5/6/13 to activate B cells towards boosted humoral immune responses. Importantly, the commitment of T cells to Th1 or Th2 cells crucially depends on the activation of redox-sensitive signaling cascades, where an oxidative environment favors the Th1 phenotype and anti-oxidative stress is prone to Th2 development [63]. It has been shown that low doses of H_2_O_2_ reduced IFNγ production by Th1 clones and increased IL4 secretion by Th2 clones [64] (Figure 2). 

T cells with varied phenotypes have different redox statuses and thus different ROS susceptibilities [65]. In general, an oxidative microenvironment exerts an opposite effect on cytokine secretion by Th1 as compared with Th2 cells [64]. It was reported that an oxidative signal originating from the mitochondrial respiratory complex I enhanced IL4 expression, which resulted in an upregulation of Th2-driven inflammation [66]. IFNγ can trigger NOX-mediated ROS formation; a study showed that T cells carrying intact NOX (an oxidase of NADPH that generates ROS) exhibited the Th1 phenotype, whereas those having mutant NOX did not after stimulation with immobilized anti-CD3 and anti-CD28 T cells in vivo [67]. Thus, increased ROS concentration is associated with an increased number of Th2 cells and a reduced number of Th1 cells. That is, a pro-oxidant environment may facilitate Th1 cell priming during the initial phase of an immune response but suppress Th1 cell proliferation if the pro-oxidant signal is sustained [65] (Figure 2). 

#### 2.2.2. Redox-Dependent TLR Control on Immune Response via Transcription Factor Activation

NFkB, a downstream target of TLRs participating in the vital pathways that activate cytokines, is a redox-sensitive dimeric transcription factor and known as a central regulator of T cell immunity. NFkB-directed signaling can increase COX2 expression on macrophages and monocytes, which results in tumor immune surveillance [54]. 

Effectors of the MAPK pathway, i.e., p38, JNK and ERK, are also known TLR targets with prominent roles in redox-regulated cell immune responses. Specifically, the MAPK-p38 pathway is activated in response to numerous cellular stimuli, including cytokines and redox stress, activation of which is required for T cell survival [68], CD4+ T cell differentiation [69], and CD8+ T cell cytokine secretion [70]. Scavenging ROS suppressed JNK activation [71], suggesting its critical role in mediating redox signaling [72], and JNK1/2 were indispensable for T cell priming into Th1 or Th2 lineages [73,74]. It has been proven that ERK has indispensable functions in the activation and proliferation of CD8+ T cells [75,76], which can be activated by a low ROS dose but suppressed by a high ROS concentration in T cells [77].

### 2.3. Steroid Hormone Receptors: Altered Metabolism

Metabolic reprogramming refers to collective alterations occurring within multiple metabolic pathways in cancer cells, with a prime example being the Warburg effect. The Warburg effect is a phenomenon by which cancer cells favor lactate production independently of the oxygen level. Specifically, healthy cells generally consume glucose through glycolysis that allows for pyruvate synthesis to fuel mitochondrial respiration towards maximal ATP production under normoxia and convert pyruvates to lactates under hypoxia (namely aerobic glycolysis); this, although fueling ATP production, is suboptimal. In the Warburg effect, cancer cells adopt aerobic glycolysis even under sufficient oxygen supply [78].

Many steroid hormone receptors have been linked to the Warburg effect, including androgen receptors (AR) [79], progesterone receptors (PR) [80], estrogen related receptors (ERRs) [78], thyroid receptor (TR) [56], and mineralocorticoid receptors (MR) [81]. Below, we selected AR and ERR to exemplify how receptors of this type contribute to cancer cell metabolic reprogramming in response to redox fluctuation.

#### 2.3.1. Androgen Receptor

As a member of the steroid hormone receptor family of the nuclear receptor superfamily, AR acts as a hormone-controlled transcription factor that relays the signals of both natural and synthetic androgens to genes and transcriptional programs. On androgen binding, AR is released from a chaperone complex in the cytosol, which results in AR homo-dimerization, nuclear translocation, interaction with androgen response elements and cofactors, and regulation of AR target gene expression.

AR positively controls several metabolism-associated pathways by regulating a vast array of transcriptional networks. Specifically, AR regulates glucose homeostasis via controlling the expression of glucose transporters and several enzymes involved in glycolysis, promoting mitochondrial respiration via regulating enzymes of the tricarboxylic acid (TCA) cycle and the electron transport chain (ETC), and controling lipid biosynthesis and fatty acid β-oxidation (FAO) [79]. 

Accumulated evidence has suggested AR as a master of metabolic reprogramming in prostate cancer cells. The metabolism of the prostate gland is unique in that glandular secretory epithelial prostate cells have a truncated mitochondrial TCA cycle to produce and secrete citrate [73,74,75,76], and AR plays a critical role in this process as castration blocks citrate secretion [77,78]. In prostate cancer cells, AR enhances aerobic glycolysis by rapidly promoting glucose uptake and usage, promoting mitochondrial respiration, and stimulating mitochondrial biogenesis. For instance, activated AR leads to a 50–450% increased glucose uptake in prostate cancer cells [82,83,84] as it can induce the level of several genes encoding glucose transporters such as SLC2A1, SLC2A3, SLC2A10, and SLC2A12 [82,85], and glycolytic genes such as HK1, HK2, PFK, PFKP, PFKFB2, and ENO1 [82,83,84]. Besides the promotive role of AR on the Warburg effect in prostate cancer cells, AR is also involved in many other processes that contribute to cancer cell metabolic reprogramming. For instance, AR increased the mitochondrial oxygen consumption rate up to two-fold following 48–72 h of androgen exposure towards restored mitochondrial function for optimal ATP synthesis [83,84,86,87,88] and enhanced de novo lipid synthesis by inducing genes associated with the key steps of this process, such as ACLY, FASN, SCD, ELOVL5, and ELOV17 [87,88,89,90,91,92,93,94,95,96,97,98,99,100,101,102,103,104,105,106,107,108,109,110,111].

AR produces ROS, and oxidative stress evokes AR signaling and contributes to the pro-survival and anti-apoptotic effects of prostate cancer cells in response to androgen deprivation [112]. For instance, redox-protective proteins such as SOD2 were reduced by androgen deprivation in castration-resistant prostate cancer cells [113,114], yet SOD2 repression contributed to the castration resistance of prostate cancer cells via AR reactivation through various mechanisms, including inducing the expression of genes involved in steroid metabolism, such as AKR1C3 [115], and genes encoding nuclear receptor co-regulators, such as NCOA4 [116]. Accordingly, SOD mimetics were proposed with therapeutic effects on prostate cancer cells by reducing oxidative stress and suppressing AR expression [117]. Also, small molecules such as A4B17 have been implicated as promising AR-positive prostate cancer therapeutics by suppressing AR target genes involved in oxidative stress and metabolism [118]. These results collectively suggested the feasibility of antioxidant agents in the treatment of hormone-associated (castration-resistant or AR-positive) prostate cancers.

#### 2.3.2. Estrogen Related Receptors

Different from AR, ERRs are orphan nuclear receptors. ERRs have three members: ERRα, ERRβ, and ERRγ [119]. Although ERRs share sequence homology with ERs, they do not bind estrogens but rather estrogen-related response elements to take on actions. For instance, ERRα could bind to specific DNA regions to modify gene expression associated with breast cancer development [120], and ERRα could form a complex with ERRγ and PGC-1α to modulate energy homeostasis in skeletal muscle cells [121].

ERRs are known as central transcriptional regulators of energy homeostasis [101,122]. They can stimulate glycolysis under normoxia by enhancing the expression of genes encoding glycolytic enzymes, such as HIF cofactors (for HIF-dependent genes), or interacting with Myc (for genes on synergistic activation of ERR and Myc), and regulate energy metabolism by orchestrating mitochondrial biogenesis, FAO, and oxidative phosphorylation (OXPHOS).

ERRs have recently been positioned as not only master regulators of cellular energy metabolism but also of ROS metabolism [123]. ERRs have both anti- and pro-oxidant effects, the regulatory direction of which is cell type- and context-specific and with varied mechanisms. Specifically, while ERRs regulate a large panel of genes encoding antioxidant enzymes, they are involved in cellular ROS production via transcriptional control of mitochondrial biogenesis and the ETC. While ERRs regulate a vast array of genes controlling cellular ROS concentration, ROS affects the activity of the ERRs by inducing thiol oxidation, which negatively regulates the binding affinity of ERRs to their ligands and targets [124,125]. Thus, enhanced understanding of the complexity of the interplay between ERRs and oxidative stress is required before novel redox-dependent therapeutic strategies can be established to take advantage of features of these family members.

## 3. Cold Atmospheric Plasma as an Emerging Redox-Modulating Tool against Aggressive Cancers via Dynamically Varying Receptor Targets

### 3.1. CAP May Resolve Drug Resistance via Targeting a Set of Dynamically Varying Receptors

While cancer cells that lack antioxidant mechanisms seen in normal cells undergo programmed cell death in response to oxidative stress, those having evolved the ability to adapt to this stress phenotype survive. Redox stress can be sensed by various types of receptors that convey the primary impact on distinct cancer hallmarks via activating different pathways. Thus, approaches interfering with cellular redox levels and targeting the ability to override redox imbalance may selectively trigger death events in cancer cells and overcome drug resistance via targeting multiple and dynamically varying receptors, which offers novel opportunities for treating aggressive cancers.

Despite the demonstrated selectivity of CAP for aggressive cancer cells, molecular pathways underpinning the response of cancer cells to CAP treatment has long been considered a black box. With intensive investigations devoted to understanding the molecular mechanisms that enable the observed selectivity of CAP against cancer cells, a common sense has been achieved on the tilted redox balance as a result of external perturbation in cancer cells that lack normal antioxidant systems, rendering malignant cells more vulnerable to ROS-triggered programmed cell death [126]. A more chemically oriented explanation that has been gaining increased popularity attributes the selectivity of CAP against cancers to the membrane feature of malignant cells. That is, cells undergoing malignant progression are characteristic of expressing NOX1, catalase, and SOD that can protect them from being attacked by exogenous reactive oxygen and nitrogen species (RONS) [127]; the unique composition of CAP can synergistically sensitize tumor cells to external redox perturbation by generating high concentrations of secondary singlet oxygen in tumor cells that inactivates catalase and promotes aquaporin-mediated H_2_O_2_ influx towards activated RONS-triggered cell death [128,129]. Other explanations include the over-represented aquaporins and low cholesterol fraction in cancer cell cytoplasmic membranes that promotes RONS permission [130,131], rendering cancer cells more sensitive to redox modulation as manifested by the selectivity of CAP against malignant tumor cells.

Various receptors have been reported to be involved in the signal relay in cancer cells in response to CAP treatment. For instance, EGFR mediated the efficacy of CAP in killing EGFR-overexpressing oral squamous cell carcinoma cells [132] and the expression of death receptors such as TNFR1 and DR4/5 were noticeably enhanced in glioblastoma cells after CAP treatment [133]. CAP is composed of various types of reactive oxygen and nitrogen species, such as hydroxyl radical (OH·), singlet oxygen (_1_O^2^), superoxide (O^2−^), nitric oxide (NO·), hydrogen peroxide (H_2_O_2_), protonated forms of peroxynitrite (ONOOH), and ozone (O^3^). While some species interact with the cancer cell surface and affect receptor-mediated signaling, some enter the cell and create redox stress. Importantly, these reactive species interact and dynamically transform into each other and create additional reactive species (Figure 3), rendering the role of CAP on cancer cells dynamically controlled by factors such as ROS concentration, as determined by CAP parameters and tumor cell surface characteristics and dictated by cell type and state. Besides, different receptors may have different sensitivities to different combinations of reactive species. Thus, CAP may function in a similar way to targeted therapy but differ in having multiple targets that vary dynamically with the concentration of reactive species and tumor cell state for a given cancer type. This may endow CAP with extreme clinical significance, as failing in the blockage of one particular pathway as a result of evolved mutations in the associated sensing receptors in cancer cells is not sufficient to disable the efficacy of CAP. In addition, by varying the treatment duration and/or dose of CAP, a dynamically varying set of targeted receptors may be obtained that guarantees its selectivity against malignant cells. These may collectively suggest a substantially reduced likelihood of developing drug resistance, a critical and intrinsic issue of targeted therapies, in CAP-based treatment modalities. Indeed, CAP has been shown capable of blocking three cancer cell survival pathways to overcome drug resistance [134]. However, additional and more focused investigations are required to validate this hypothesis towards our advanced understandings of possible benefits that CAP can bring to cancer management.

### 3.2. Challenges Hindering the Clinical Translation of CAP as an Onco-Therapy

As a promising onco-therapeutic, alone or combined with other treatment modalities, the clinical translation of CAP for cancer control is still at its initial stage. The first clinical endeavor was made by Dr. Keith Millikan, whose use of CAP expanded the life span of a 75-year-old patient carrying late-stage pancreatic cancer for two additional years [135,136]. The first clinical trial using CAP as an oncotherapy (NCT04267575) was approved in 2019 by the FDA and carried out by US Medical Innovations, where 17 out of the 20 solid malignant tumor patients recruited were still alive by the completion of this study [135]. These results collectively suggested the safety and clinical efficacy of CAP in cancer treatment.

Though CAP has showcased its great promise as an onco-therapy in clinics, it is worth mentioning that the biological outcome of CAP is dose-dependent [137] and may vary with the tumor type or cancer state [138]. Besides CAP-triggered apoptosis that has been widely reported, CAP was shown capable of inducing autophagy in human melanoma cells [139,140]; arresting cells at the G0/G1 stage in AR-independent prostate cancer cells [141]; inducing ferroptosis in human lung cancer cells [142] and malignant mesothelioma cells [143]; causing senescence in melanoma cells [144]; and triggering immunogenic cell death (ICD) in melanoma cells [145], A549 lung carcinoma cells [146] in vitro, and in murine CT26 colorectal tumors in vivo [147]. Importantly, the sensitivity of cancer cells to redox modulation differs significantly. Taking prostate cancers as an example, while AR-independent prostate cancer cells are sensitive to CAP-induced ROS elevation [141], antioxidant treatment is feasible in AR-positive or castration-resistant prostate cancers [118]. These evidences suggest that although the dose-dependent feature of CAP largely improves its flexibility in cancer treatment that may prevent malignant cells from developing drug resistance, it also creates complexity in determining the optimal dose for a particular patient. Thus, establishing a CAP-based diagnostic and therapeutic platform that can accurately locate the dose and medication strategy (e.g., CAP alone or being combined with other agents; treatment frequency; therapeutic methodology) of CAP for an individual patient becomes of particular importance and challenges its clinical translation. Receptors are likely to play a significant role in this delineation.

Lastly, besides the effect on cancer cells, determining whether and how CAP affects other tissues such as extracellular matrices (ECM) and collagens requires intensive investigations before CAP can be safely translated into clinics as a onco-therapeutic approach. For instance, CAP was shown to influence cellular behaviors towards accelerated or suppressed chondrogenesis and endochondral ossification [148] and modify dentin collagen through the crosslinking effect [9]. Thus, determining how to maximize the onco-therapeutic efficacy of CAP while minimizing these potential side effects via selectively activating a panel of receptors may represent a promising solution. 

## 4. Conclusions

As the primary sensors of cells to the external environment, receptors largely mediate cell responses to external stimuli perturbation and offer a lens through which to delineate precise molecular mechanisms of CAP in cell redox level manipulation and cancer treatment. In this way, cell surface receptors largely characterize cell responsiveness to CAP in modulating cellular redox imbalance, determining the spectrum of cancers feasible for receiving CAP treatment, and explaining the anti-cancer efficacy of this emerging redox modulating tool for aggressive cancers. Importantly, the demonstrated efficacy of CAP in selectively halting cancer cell proliferation and migration, triggering immune response, and reprogramming cell metabolism can be attributed to a differential combination of receptor-mediated cell signaling in response to CAP-triggered redox fluctuation and toxicity. Thus, CAP differs from canonical targeted therapies in having multiple and dynamically varying sets of targeted receptors, the dominance of which is determined by factors such as the dose and treatment strategy of CAP, as well as the type and disease course of the tumor. Dysregulation of these cancer attributes collectively results in the very high anti-cancer capacity of CAP and significantly reduces the rate at which cancer cells idevelop drug resistance after receiving CAP treatment. Despite the promising contribution of CAP to the cancer cure, less focus has been put on the role of receptors in mediating CAP’s selectivity against malignant cells, the emphasis of which may result in the clarification of this black box (i.e., CAP) from the ‘receptor’ perspective and should lead to one important future trend. This also adds complexity to the clinical use of CAP as a precision onco-therapy, where the establishment of a theranostic system capable of precisely diagnosing the dose of CAP for a particular patient and suggesting the treatment strategy accordingly becomes imperative and warrants intensive attention.

## Figures and Tables

**Figure 1 biomolecules-12-01880-f001:**
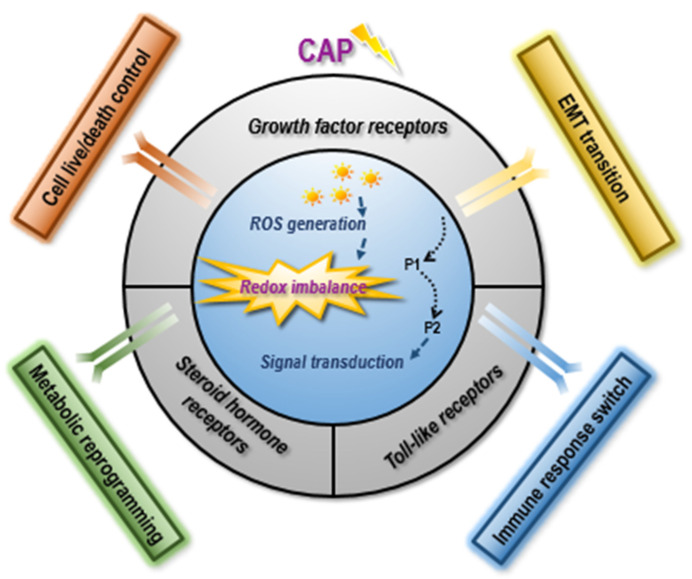
Conceptual illustration on the association of primary receptor groups with critical carcinogenesis events in response to redox imbalance. The 10 cancer hallmarks can be matched to four critical transition events during carcinogenesis: cell live/death control, epithelial-to-mesenchymal transition (EMT), immune response switch, and metabolic reprogramming [10]. Cell live/death and EMT are largely controlled by signaling involving growth factor receptors, the immune response switch is tightly associated with Toll-like receptors, and metabolic reprogramming is closely linked to steroid hormone receptors, among others, in response to redox imbalance. CAP, by imposing redox stress in cells and creating redox imbalance in cancer cells, can selectively halt cancer cell growth and EMT, induce anti-tumor immune response, and rewire altered metabolism in aggressive cells. Given the redox nature of CAP, its roles in arresting cancer hallmarks, and the impact of redox imbalance on cancer cells, it is likely that CAP conveys its selectivity against cancer cells by targeting multiple cell receptors in a dynamically varying manner that needs in-depth exploration.

**Figure 2 biomolecules-12-01880-f002:**
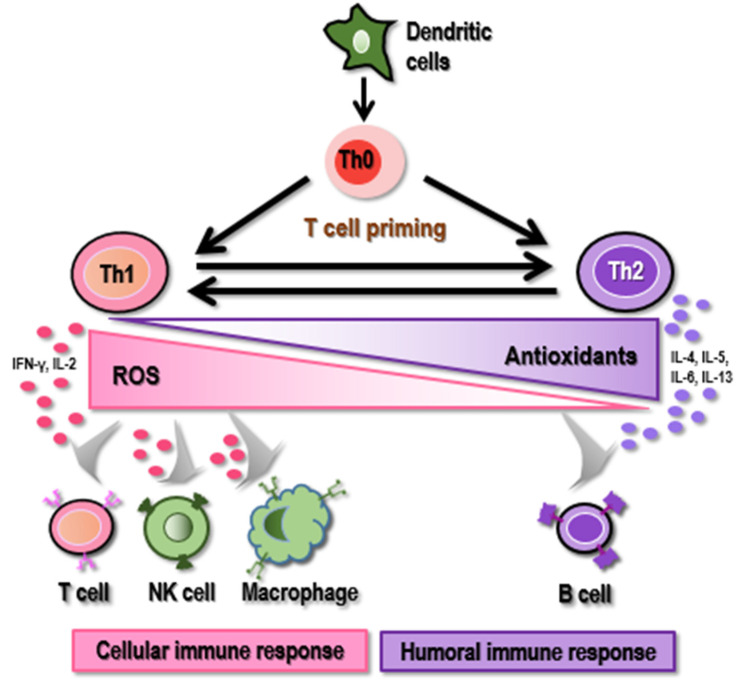
Impact of redox level on the priming of T helper cells. High ROS level favors the commitment of CD4+ T cells (Th0) to Th1 cells that secrete pro-inflammatory cytokines such as IFNγ and IL2 to stimulate cytotoxic T cells, NK cells, and macrophages and switch on cellular immune responses. Low ROS level triggers the priming of Th0 cells to Th2 cells that release IL4/5/6/13 to activate B cells towards boosted humoral immune responses.

**Figure 3 biomolecules-12-01880-f003:**
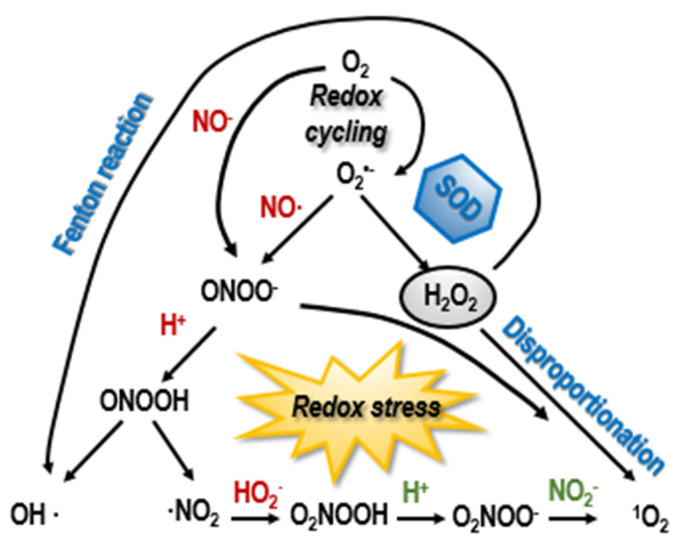
Illustrative diagram showing interactions and creations of reactive species. CAP is composed of various types of reactive oxygen and nitrogen species, including hydroxyl radical (OH•), singlet oxygen (_1_O^2^), superoxide (O^2−^), nitric oxide (NO•), hydrogen peroxide (H_2_O_2_), and protonated forms of peroxynitrite (ONOOH). These species dynamically interact to transform into each other or create new species, the process of which is determined, under the context of CAP-cell interactions, by CAP ejection parameters and cell surface traits. In this diagram, ‘red’ colored species are added to a reaction, and ‘green’ colored species are removed from the corresponding reactant.

## Data Availability

Not applicable.

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
