# Peer review of "Receptor-Mediated Redox Imbalance: An Emerging Clinical Avenue against Aggressive Cancers"

_biomolecules, 2022, doi:10.3390/biom12121880_

Round 1

Reviewer 1 Report

The authors should give a comprehensive literature review in the introduction.   

Author Response

Dear editors,

We are pleased to resubmit our revised manuscript, namely ‘Receptor-mediated redox imbalance: an emerging clinical avenue against aggressive cancers’, for your consideration towards its publication.

We thank all reviewers for these pertinent suggestions that have helped us sufficiently improved this paper, and have addressed all these comments. Please let us know whether anything additional is needed.

BR,

Xiaofeng Dai

Wuxi School of Medicine

Jiangnan University

Reviewer 2 Report

Authors have taken a unique topic in redox imbalance by cancer cell receptors. 

Comment: Authors are requested to included in the role extracellular matrix and following with collagen in  the session of 3.3 Challenges hindering the clinical translation of CAP as an onco-therapy. 

Author Response

(The authors gave the same response as above.)

Reviewer 3 Report

This review is well written. I would recommend publication after the authros addressed the following 2 minor questions.

1. Line 50-52: "The 10 cancer hallmarks can be matched to four critical transition events during 50 carcinogenesis, i.e., cell live/death control, epithelial-to-mesenchymal transition (EMT), immune response switch, and 51 metabolic reprogramming."

If possible, please add references to the above statement. 

2. A recent publication which is important in unraveling the mechanism of plasma redox modulation is worth mentioning in this review: Miebach L, Freund E, Clemen R, et al. Gas plasma–oxidized sodium chloride acts via hydrogen peroxide in a model of peritoneal carcinomatosis[J]. Proceedings of the National Academy of Sciences, 2022, 119(31): e2200708119.

Author Response

(The authors gave the same response as above.)
